# *Culex quinquefasciatus* Resistant to the Binary Toxin from *Lysinibacillus sphaericus* Displays a Consistent Downregulation of Pantetheinase Transcripts

**DOI:** 10.3390/biom14010033

**Published:** 2023-12-25

**Authors:** Tatiana M. T. Rezende, Heverly S. G. Menezes, Antonio M. Rezende, Milena P. Cavalcanti, Yuri M. G. Silva, Osvaldo P. de-Melo-Neto, Tatiany P. Romão, Maria Helena N. L. Silva-Filha

**Affiliations:** 1Department of Entomology, Instituto Aggeu Magalhães-Fiocruz, Recife 50740-465, PE, Brazil; tatianateo@gmail.com (T.M.T.R.); heverly.menezes@fiocruz.br (H.S.G.M.); yurimateusgarcia@gmail.com (Y.M.G.S.); tatiany.romao@fiocruz.br (T.P.R.); 2Department of Microbiology, Instituto Aggeu Magalhães-Fiocruz, Recife 50740-465, PE, Brazil; antonio.rezende@fiocruz.br (A.M.R.); milena.paiva@fiocruz.br (M.P.C.); osvaldo.pompilio@fiocruz.br (O.P.d.-M.-N.); 3National Institute for Molecular Entomology, Rio de Janeiro 21941-902, RJ, Brazil

**Keywords:** transcriptome, RNA-seq, vanin-like protein, receptor, Cqm1, pesticidal toxins, microbial larvicides

## Abstract

*Culex quinquefasciatus* resistance to the binary (Bin) toxin, the major larvicidal component from *Lysinibacillus sphaericus*, is associated with mutations in the *cqm1* gene, encoding the Bin-toxin receptor. Downregulation of the *cqm1* transcript was found in the transcriptome of larvae resistant to the *L. sphaericus* IAB59 strain, which produces both the Bin toxin and a second binary toxin, Cry48Aa/Cry49Aa. Here, we investigated the transcription profiles of two other mosquito colonies having Bin resistance only. These confirmed the *cqm1* downregulation and identified transcripts encoding the enzyme pantetheinase as the most downregulated mRNAs in both resistant colonies. Further quantification of these transcripts reinforced their strong downregulation in Bin-resistant larvae. Multiple genes were found encoding this enzyme in *Cx. quinquefasciatus* and a recombinant pantetheinase was then expressed in *Escherichia coli* and *Sf*9 cells, with its presence assessed in the midgut brush border membrane of susceptible larvae. The pantetheinase was expressed as a ~70 kDa protein, potentially membrane-bound, which does not seem to be significantly targeted by glycosylation. This is the first pantetheinase characterization in mosquitoes, and its remarkable downregulation might reflect features impacted by co-selection with the Bin-resistant phenotype or potential roles in the Bin-toxin mode of action that deserve to be investigated.

## 1. Introduction

Microbial larvicides based on *Lysinibacillus sphaericus* and *Bacillus thuringiensis* serovar. *israelensis* (Bti) have been effectively employed for the control of different mosquitoes, including species of medical importance [1,2]. The main active ingredient found in these bacteria are protoxins found within crystal inclusions that have a selective mode of action. Upon ingestion by the insect larvae, these crystals are solubilized in the alkaline pH of the mosquito midgut and release the protoxins into the lumen [3]. The protoxins are then proteolytically processed into toxins that specifically bind to midgut receptors. The toxic effect of the commercially available *L. sphaericus* larvicides is mainly based on the crystal containing the Binary (Bin) protoxin [3,4]. The Bin toxin receptor in *Culex quinquefasciatus,* named Cqm1, is an α-glucosidase which is bound to the membrane of midgut cells through a glycosylphosphatidylinositol (GPI) anchor [5]. Some *L. sphaericus* strains can produce other mosquitocidal toxins that are currently under study but are not yet used as active ingredients of larvicides, such as a second Cry48Aa/Cry49Aa binary toxin [6,7]. The insecticidal crystal produced by Bti is more complex and consists of four major protoxins (Cry4Aa, Cry4Ba, Cry11Aa, and Cyt1Aa) [3] that act in synergy [8]. The Cry toxins need to bind first to Cyt1Aa, which is inserted within the cell membrane, as a precursor step. The Cry–Cyt1Aa interaction then induces conformational changes that allow the Cry toxins to further bind with high affinity to different classes of larvae midgut receptors [9,10,11].

Microbial larvicides containing crystals from *L. sphaericus* and Bti are effective tools used as part of integrated mosquito control programs, due to their high larvicidal action associated with environmental safety and an overall low risk of resistance selection [12,13,14,15]. Resistance to the Bin toxin has nevertheless been reported in *Cx. quinquefasciatus* and it remains a major concern. It can be caused by the loss of functional toxin receptors within the mosquito midgut, leading to resistance levels exceeding 1000-fold [16]. The Bin toxin is a heterodimer formed by the BinA and BinB subunits acting in synergy [17,18]. The BinB component recognizes and binds to the Cqm1 receptor, and both subunits are internalized by the insect cells where only BinA is responsible for the toxic effects [19]. The Bin action can then be disrupted by its failure to bind to its receptor or by the absence of the receptor from the insect midgut [16,20]. Indeed, studies have identified mutations in *cqm1* alleles that result in the production of truncated Cqm1 variants lacking the GPI anchor and that have a much-reduced overall expression [21,22,23,24]. The GPI anchor is critical for the receptor localization on the membrane of the midgut epithelium and an impaired localization, or a reduced expression of the receptor, prevents binding of the BinB subunit to the larvae midgut.

Our laboratory maintained and studied three Bin-resistant *Cx. quinquefasciatus* colonies (RIAB59, REC, and REC-2) whose larvae are homozygous for different *cqm1* alleles that confer Bin resistance and are associated with a recessive inheritance [21,25,26]. RIAB59 is a mosquito colony highly resistant to the Bin toxin (RR > 1000-fold) that is also moderately resistant (RR ≈ 15-fold) to the Cry48Aa/Cry49Aa toxin [26]. Bin is the major insecticidal component of the *L. sphaericus* IAB59 strain, which also produces the Cry48Aa/Cry49Aa toxin [6,7]. A transcriptome analyses comparing RIAB59 larvae with larvae from a susceptible colony revealed a profile of differentially expressed genes, including 57 downregulated and upregulated transcripts exhibiting log2 fold-changes greater than three [26]. The *cqm1* transcript was the fourth most downregulated gene, consistent with the homozygous genotype for the *cqm1_REC_* resistant allele found in the RIAB59 larvae [25]. Notably, transcripts encoding the enzyme pantetheinase, belonging to the family of vanin-like proteins, were identified among the most downregulated in the resistant colony [26]. This protein was also found as a potential ligand for the Cry48Aa/Cry49Aa toxin in a previous proteomic analysis, along with other midgut microvilli proteins from *Cx quinquefasciatus* larvae [27]. Pantetheinases have not yet been studied in mosquitoes, while they have been widely investigated in humans, displaying functions related to the metabolism of lipids, oxidative stress, and immunity [28]. Here, we report on further transcriptomic analyses focusing on two *Cx quinquefasciatus* colonies which are resistant only to the Bin toxin (REC, REC-2). Our findings reinforce the evidence implying a selection for lower pantetheinase expression linked to the resistance phenotype to *L. sphaericus*. We thus sought to investigate the pantetheinases further in *Cx. quinquefasciatus* and to perform an in vitro identification through recombinant expression in a heterologous system.

## 2. Materials and Methods

### 2.1. Culex quinquefasciatus Colonies

In this study, two *Cx quinquefasciatus* colonies resistant to the *L. sphaericus* strain 2362 (REC, REC-2), which contains the Bin toxin, and the susceptible colony (CqSLab), referred hereafter to as S, were investigated. These colonies were maintained at the insectary of the Instituto Aggeu Magalhães (IAM)-Fiocruz under controlled conditions of 26 ± 1 °C, 70% of relative humidity, and a photoperiod of 14 h:10 h of light:darkness. Larvae were reared in tap water and fed cat food (Whiskas^®^, Ribeirão Preto, SP, Brazil). The adults were fed sucrose solution (10%), and females were also fed defibrinated rabbit blood once a week. The S colony was established with eggs collected in the city of Recife, Brazil, and has been maintained for more than eleven years, as a reference colony susceptible to insecticides [21]. The REC and REC-2 colonies display resistance ratios (RR) to the Bin toxin of ~4000-fold, attributed to the homozygous genotype for the *cqm1_REC_* and *cqm1_REC-2_* resistance alleles, respectively [29]. Both Bin-resistant colonies have been continuously maintained and have been exposed to *L. sphaericus* 2362 at every five generations. Additional information about the establishment and features of the REC and REC-2 strains is available in Chalegre et al. [21].

### 2.2. RNA-Seq Assays

Three pools of twenty midguts from early fourth-instar larvae were used for RNA extraction from each colony, using the RNeasy Mini Kit (Qiagen, Hilden, Germany). The integrity of the RNA was assessed through agarose gel electrophoresis and the purity and quantification of these samples were determined using a NanoDrop 2000™ spectrophotometer (Thermo Fisher Scientific, Waltham, MA, USA) and Qubit 2.0 Fluorometer (Thermo Fisher Scientific). Preparation of paired-end libraries was carried out using total RNA and the TruSeq Stranded mRNA Library Prep kit (Illumina, San Diego, CA, USA). The libraries were then sequenced on an Illumina MiSeq Sequencer using the MiSeq™ Reagent Kit V3 150 cycles (Illumina). The analyses of the sequence results were performed according to Rezende et al. [26], using only bases with Phred scores higher than 30, and the libraries mapped with the genome assembly of the *Culex quinquefasciatus* Johannesburg strain, CpipJ2 (file: culex-quinquefasciatus-johannesburgscaffoldscpipj2.fa) available at the VectorBase database (https://www.vectorbase.org accessed on 1 February 2018). Mapping was performed using the STAR aligner v.2.5.0 [30] with default parameters, except for the quantMode option. The gene count data were then imported into the R environment and subsequently transformed to the log2 scale, using the rlog transformation function from the DESeq2 package [31], version 1.40.2. The data were then used as input for the prcomp function to perform the principal component analysis (PCA). The function *plot* from R environment was then applied to plot the first and second principal components calculated from each sample. Differential expression analyses were performed using the DESeq2, considering only genes with a minimum representation of five reads across all three biological replicates and in at least one condition (resistant or susceptible). Genes that exhibited an absolute log2 fold change equal to or greater than 1, and false discovery rates (FDR) corrected *p*-values lower than 0.05, were selected for further investigation. The correlation between corrected *p*-values and log2 fold change was visualized using Volcano plots generated using the EnhancedVolcano package, version 1.18.0. After annotation using Vectorbase, the most prominent DEGs was manually revised using UniProt data and Blastx (https://www.uniprot.org accessed on 6 November 2023). Functional enrichment analyses of DEGs with Gene Ontology (GO) terms and InterPro protein domain descriptions were performed with the DAVID tool (https://david.ncifcrf.gov accessed on 6 November 2023).

### 2.3. RT-qPCR Assays

Quantitative reverse transcription polymerase chain reaction (RT-qPCR) assays were performed to validate the abundance of selected DEGs identified through the RNA-seq in larvae from the S, REC, and REC-2 colonies. For this validation, the same RNA samples used for the RNA-seq assays, from pools of midguts from twenty larvae, were evaluated. RT-qPCR was also used to investigate the profile of the pantetheinase transcript from individual larvae, with total RNA extracted from each larva (*n* = 30–45 larvae/colony) using Trizol™ reagent (Thermo Fisher Scientific), according to the manufacturer’s instructions. The QuantiTect^®^ SYBR Green RT-PCR^®^ Kit (Qiagen) was used to perform the RT-qPCR reactions, following the manufacturer’s instructions. Specific primers for the target genes and for the ribosomal protein *18S,* used as the endogenous control gene, are described in Appendix A. These primers were verified using Blast service against reference RNA sequences for *Cx. quinquefasciatus*. The reaction results were analyzed using the QuantStudio^®^ 5 System (Thermo Fisher Scientific) and the relative quantification was performed using the Applied Biosystems^TM^ Analysis software v.2.3 with the Relative Quantification Analysis Module v.3.3. For the validation of the DEGs, the means and standard errors from three biological replicates from each colony were determined. For the relative quantification of the pantetheinase transcripts in individual larvae, the cycle threshold (CT) of one S larva, representing the average of the CT values found in the replicates, was used as the reference sample [32]. These analyses were performed with GraphPad Prism v.9.5.1 for Windows (GraphPad Software Inc., San Diego, CA, USA), using the Student’s *t*-test and considering a *p*-value < 0.05 to be statistically significant.

### 2.4. In Silico Analyses

A comparative analysis of the amino acid sequences of *Cx. quinquefasciatus* pantetheinases available at Vectorbase (www.vectorbase.org accessed on 20 September 2023) was conducted using the Bioedit v.7.2.5 for Windows (https://thalljiscience.github.io accessed on 20 September 2023). Multiple sequence alignments were performed using the clustalW program, and based on the BLOSUM62 Matrix, with amino acids that were identical in more than 60% of the sequences highlighted in black and amino acids considered similar on more than 60% of the sequences, shown in gray. In silico analyses of putative post-translational modifications on the pantetheinase sequences were carried out to evaluate the presence of N-glycosylations, O-glycosylations, and GPI-anchor sites, using the NetNGlyc 1.0 (https://services.healthtech.dtu.dk/service.php?NetNGlyc-1.0, NetOGlyc 4.0 (https://services.healthtech.dtu.dk/service.php?NetOGlyc-4.0, and the big-PI (https://mendel.imp.ac.at/gpi/gpi_server.html predictors, respectively, accessed on 10 January 2023.

### 2.5. Cloning and Expression Procedures in Escherichia coli and Sf9 Cells

To clone the gene encoding the selected *Cx. quinquefasciatus* pantetheinase (CPIJ017593), a reverse transcription reaction was set up using RNA extracted with Trizol from a pool of ten fourth-instar larvae from the S colony. The cDNA synthesis was carried out using the SuperScript ^®^ III First-Strand Synthesis System kit (Thermo Fisher Scientific), followed by PCR reactions using primers designed according to the gene sequence (GenBank accession no. EDS45922.1) and including sites for the *EcoR*I and *Xho*I restriction enzymes (forward 5′-CGAGAATTCATGAGGATCGTCTTGGCG-3′; reverse 5′-CAACTCGAGAGGATCTTGATCCTGGAC-3′), with restriction sites underlined. The cloned genomic fragment is missing the last 48 pb of the 1551 pb coding sequence, encoding the C-terminal 16 residues of the protein and any eventual GPI anchor. The amplified product was cloned into the *EcoR*I/*Xho*I sites of the pET21 prokaryotic vector, which adds a C-terminal poly-histidine tag to the encoded protein. The resulting plasmid (*pan*-pET21) was transformed into *E. coli* cells (BL21 star). Recombinant protein expression in *E. coli* was induced in Luria-Bertani medium supplemented with 0.5 mM IPTG. After lysis by sonication, protein purification was performed using the Ni-NTA resin^®^ (Qiagen), according to the manufacturer’s protocol. The purification yield was analyzed by SDS-PAGE and immunodetection performed with a monoclonal anti-histidine antibody. The identity of the protein was also determined by mass spectrometry (LC-MS/MS) coupled to liquid chromatography in the Orbitrap Fusion Lumos system in the facility at the Instituto Carlos Chagas-Fiocruz. To produce rabbit polyclonal antibodies, the purified protein (1.2 mg) was separated in a 12% preparative SDS-PAGE gel, with the corresponding pantetheinase band excised and sent for anti-serum production by the CélulaB company (https://www.ufrgs.br/celulab). The resulting total immune antiserum was further used for affinity purification of the anti-pantetheinase antibodies, as previously described [33]. For the expression in the ovarian cell line derived from the lepidopteran *Spodoptera frugiperda* (*Sf*9), the 1503 bp gene fragment was reamplified with primers flanked by sites for the *Kp*nI/*Xb*aI restriction enzymes (forward 5′-CAGGGTACCATGAGGATCGTCTTGGCG-3′; reverse 5′-CAGTCTAGAGGATCTTGATCCTGGACAAA-3′) and cloned into the *Kp*nI/*Xb*aI sites of the pIZT/V5-His^®^ expression plasmid (Thermo Fisher Scientific). *Sf*9 cells were transfected with the resulting plasmid (*pan*-pIZT/V5-His) using the cellfectin™ II reagent (Thermo Fisher Scientific) and the InsectSelect™ Glow System Kit (Thermo Fisher Scientific), according to the manufacturer’s instructions. The cell cultures were then selected with Zeocin™ (150–300 μg/mL) (Thermo Fisher Scientific), added 48 h after transfection, and stable cell lines were then generated. The identity of the *Sf*9 recombinant protein was also confirmed by LC-MS/MS.

### 2.6. Preparation of Protein Samples

The recombinant proteins from *E. coli* cultures or from the confluent *Sf9* cells culture were, respectively, purified using the Ni-NTA resin^®^ or harvested through precipitation of 1 mL media samples with 10% trichloroacetic acid (TCA). The protein content in the Ni-NTA eluates were determined by a comparative gel assay using a standard curve of bovine serum albumin. The native proteins from midgut brush border membrane fractions (BBMF) from *Cx. quinquefasciatus* fourth-instar larvae were prepared using either dissected midguts [4] or whole larvae [34]. The enrichment of the α-glucosidase activity (EC 3.2.1.20) in the BBMF samples was used as a marker for proteins from the apical cell membranes [21]. All samples were stored at −70 °C until use.

### 2.7. Immunodetection and Deglycosylation Assays

For the immunodetection assays, the protein samples separated by 10% SDS-PAGE were transferred to Protam Premium NC™ nitrocellulose membranes (Amershan, Little Chalfond, UK) which were then subjected to blocking in TBS-T (50 mM Tris-HCl/150 mM NaCl/0.1% Tween 20 pH 7.6), supplemented with 5% dry milk, and used for immunodetection using standard procedures [5]. For the detection of the histidine-tagged protein, a mouse monoclonal anti-His antibody (#H1029, Sigma-Aldrich, St. Louis, MO, USA) was used at a 1:5000 dilution followed by incubation with the secondary anti-mouse IgG, conjugated with horseradish peroxidase, at a 1:10,000 dilution. Alternatively, the affinity purified anti-pantetheinase antibodies, purified from the rabbit polyclonal serum, were used at a 1:100 dilution followed by incubation with the anti-rabbit IgG secondary antibody, conjugated to horseradish peroxidase, at a 1:10,000 dilution. Chemiluminescence detection was performed using the Immobillon^®^ Forte Western HRP Substrate (Millipore, Billerica, MA, EUA) and the signal was recorded using the iBright™ Imaging System (Thermo Scientific). To investigate the presence of N-glycans linked to the proteins, endoglycosidase assays were carried out with 20 μg samples of previously denatured recombinant pantetheinase incubated with PNGase F (#P0704S, New England Biolabs, Beverly, MA, USA) at 37 °C for 1 to 4 h, according to the manufacturer’s instructions. In a second procedure, nondenatured samples of the recombinant protein were incubated at 37 °C during 16 h. Protein samples incubated under the same conditions but without PNGase F were used as untreated controls. All samples were subjected to 10% SDS-PAGE and visualized through Coomassie blue staining.

## 3. Results

### 3.1. Transcriptomic Profile of Bin-Resistant Cx. quinquefasciatus Colonies

RNA samples from the midgut of *Cx. quinquefasciatus* larvae from two Bin-resistant (REC, REC-2) and the susceptible (S) colonies were subjected to whole transcriptome shotgun sequencing to identify differentially expressed genes (DEGs). For that purpose, each resistant colony was compared with the reference S colony, with the sequencing and libraries dataset available in Appendix A. A total of 14.25 × 10^6^ reads were mapped for the REC × S comparison and 15.29 × 10^6^ reads for REC-2 × S, representing at least 98% of the total number of reads sequenced. These reads were mapped against the *Cx. quinquefasciatus* genome, and the unique mappings accounted, on average, for 52.3% and 67.8% of the reads in the resistant and susceptible colonies, respectively. Analyses of the genes that fulfilled the threshold parameters, with a minimum representation of five reads for all three biological replicates in at least one condition, resistant or susceptible, revealed 5633 genes for the REC × S and 6464 genes for the REC-2 × S comparison, corresponding to approximately 28% and 32% of the annotated *Cx. quinquefasciatus* genome, respectively. Principal component analyses (PCA) of the dataset demonstrate clear segregation profiles when samples from the two resistant colonies are compared with those from the susceptible colony (Appendix A). The PCA plots suggest that there are substantial differences in the overall transcriptional profiles for the resistant colonies. To gain a deeper understanding of the molecular changes underlying these differences, we performed differential gene expression analyses. These led to the identification of several hundred DEGs with absolute log2 fold changes (log2FC) equal to or exceeding 1 and adjusted *p*-values lower than 0.05.

The analyses of the DEGs found comparing each resistant colony to the single S colony revealed 610 downregulated and 494 upregulated genes for the REC colony, while for REC-2 there were 517 downregulated and 402 upregulated genes (Appendix A). These genes, highlighted by volcano plots (Figure 1), are likely associated with the biological processes driving the differences between the colonies. For a first assessment of functionality, DEGs found with each of the resistant colonies were then classified according to the Gene Ontology (GO) of enriched terms in the Cellular Components (CC), Biological Process (BP) or Molecular Function (MF) categories, as well as with terms related to the identified Interpro (IP) domains (Appendix A). The GO terms grouping DEGs from the REC × S comparison include one term for the CC category (endoplasmic reticulum membrane); two terms for BP (carbohydrate and glutathione metabolic processes); and five terms for MF (alpha-amylase activity plus chitin, heme and iron binding, and monooxygenase and oxidoreductase activities). As for the IP analysis, it showed gene enrichments in several terms of detoxifying enzymes commonly involved in the metabolism of xenobiotics, which include the chemical insecticides. The DEGs from the REC-2 × S comparison were grouped into three GO terms from the CC category (intracellular membrane-bounded organelle, extracellular space, and region); one term from BP (glutathione metabolic process), also found for the REC × S analysis; and six terms for MF, with five of those also found for the REC × S comparison (chitin, heme and iron binding, plus monooxygenase and oxidoreductase activities) and also the term related to alpha-amylase activity. A broader profile of IP terms was identified for the REC-2 × S comparison (all six terms found for the REC × S analysis plus several others), but, in general, it also showed an abundance in terms related to resistance genes associated with insecticide metabolism. To date, however, there is no report on the involvement of any of these protein families in the metabolism of the Bin toxin in mosquitoes.

### 3.2. Analysis of DEGs Found When Comparing the Bin-Resistant Colonies

For a comparative analysis of the most downregulated or upregulated transcripts found in the Bin-resistant larvae, those found in Appendix A with log2FC ≥ 3.5 were used to generate the Venn diagrams shown in Figure 2. The list of most downregulated DEGs for the REC colony consisted of 38 transcripts, with 17 of those also found among the most downregulated transcripts from the REC-2 larvae (Figure 2). Eleven other transcripts were found among those most downregulated with the REC-2 larvae only. For the most upregulated DEGs, eight were common to both colonies, while six and 13 transcripts were found among those most upregulated only with the REC or REC-2 colonies, respectively. The *cqm1* transcript (CPIJ013173) was one of the most downregulated DEGs found with both the REC (log2FC = −5.84) and REC-2 larvae (log2FC = −4.14). This result is consistent with the previous description of Cqm1 as the Bin-toxin receptor and with the substantial reduction in its abundance previously reported from the midgut cells of Bin-resistant larvae [21], therefore validating the transcriptomic approach carried out here. It is worth noting that more transcripts are found in common to be differentially expressed for both REC and REC-2 larvae, with DEGs having a log2FC ≥ 3.5 in one colony but with a log2FC in the second colony ≤ 3.5 but still greater than 1. Despite not being included in the lists from Figure 2, they are indicated in both diagrams. For this analysis, we also considered our previously published data on DEGs found associated with the RIAB59 colony, since it consists of individuals also resistant to the Bin toxin [26]. Nine transcripts were then found to be substantially downregulated in all three colonies, with two transcripts also upregulated in all. The CPIJ017593 pantetheinase transcript was the topmost downregulated in REC colony and the second most downregulated in the REC-2 larvae, with log2FC values greater than eight in both comparisons. Another pantetheinase gene (CPIJ017592) was the most downregulated transcript for the REC-2 colony (log2FC = 8.27) and was the fourth in the REC larvae (log2FC = 7.87). Most relevantly, the same pantetheinase transcripts were also the most downregulated DEGs reported for the RIAB59 colony [26].

The list of most downregulated DEGs found for both resistant colonies also included several other transcripts that might be investigated for their potential relevance to the resistance phenotype, such as those coding for ankyrins (CPIJ018744, CPIJ013539), proteins that link membrane proteins to the cytoskeleton [35]; a chitin-binding cuticle protein (CPIJ014435) and regulcalcin (CPIJ007230), reported for the uptake of insect hexamerin storage proteins from hemolymph [36], both related to insect metamorphosis and development; glucosyl/glucuronosyl transferases (CPIJ003695) and CHKov 1 (CPIJ012700), related to the processes of xenobiotic detoxification [37,38]; and CHK kinase-like domain-containing proteins (CPIJ012697, CPIJ012702), which play roles in cell cycle control and regulator of the DNA damage in response to cell injuries [39]. Among the most upregulated transcripts found for both colonies there were two that encode proteins also involved in metamorphosis: arylphorinin (CPIJ009033), which plays an important role in insect development [40,41]; and farnesol desydrogenase, responsible for the catalysis of farnesol into farnesal, a step essential for the synthesis of the insect juvenile hormone [42].

### 3.3. Validation of the RNA-Seq Transcription Status

The differential transcription statuses of five DEGs revealed by RNA-seq in the REC and REC-2 larvae were validated through RT-qPCR assays with the same RNA samples also used for the RNA-seq experiments. The relative expression levels of the *cqm1* gene transcript showed a significant reduction in the resistant larvae, when compared with the larvae from the susceptible colony, with log2FC values of −9.63 (t_(4)_ = 23.27; *p* = 0.0001) and −5.41 (t_(4)_ = 12.99; *p* = 0.0002) for the REC and REC-2 larvae, respectively (Figure 3A). A significant downregulation was observed for the selected pantetheinase gene, CPIJ017593, in both resistant colonies with log2FC values of −2.74 (t_(4)_ = 5.293; *p* = 0.0061) for REC and −4.09 (t_(4)_ = 4.726; *p* = 0.0091) for REC-2, when compared to the S colony (Figure 3B). Another downregulated transcript identified in the RNA-seq analysis, encoding the sodium/chloride-dependent amino acid transporter (CPIJ012066), also displayed significantly lower expression levels in both resistant colonies with log2FC of −9.96 (t_(2)_ = 42.59; *p* = 0.0006) for REC and −9.29 (t_(3)_ = 20.47; *p* = 0.0003) for REC-2 (Figure 3C). The upregulation of two genes identified in the RNA-seq analysis, the carboxypeptidase B precursor (CPIJ010801) and farnesol dehydrogenase gene (CPIJ002522), was also confirmed. For carboxypeptidase B precursor the log2FC values were 5.16 (t_(4)_ = 10.49; *p* = 0.0005) for REC and 3.97 (t_(4)_ = 41.56; *p* = 0.0001) for REC-2, while for farnesol dehydrogenase these were 7.99 (t_(3)_ = 22.39; *p* = 0.0002) for REC and 5.55 (t_(3)_ = 9.667; *p* = 0.0024) for REC-2 (Figure 3D,E). These results support the differential expression pattern observed by the RNA-seq, including the downregulation of the pantetheinase transcripts in larvae from the two Bin-resistant colonies. The full data from those assays are available in Appendix A.

### 3.4. In Silico Investigation on the Cx. quinquefasciatus Pantetheinase Genes

The consistent downregulation of the pantetheinase transcripts in different Bin-resistant colonies prompted us to select the corresponding protein encoding genes for further investigation. In silico analysis based on the Vectorbase dataset revealed six paralogs in *Cx. quinquefasciatus* encoding pantetheinase variants, all arranged into two sets of contiguous genes (Appendix A) which are schematically described in Figure 4. Those genes are all currently annotated as pantetheinases or vanin 1-like precursors, except for the CPIJ007584 sequence, annotated as an uncharacterized protein. The CPIJ017592 and CPIJ017593 genes are positioned next to each other and encode proteins sharing 62% identity, with 74% identity seen for the corresponding DNA sequences. These gene sequences are characterized by distinct exon/intron arrangements, with four and three introns found within their coding regions, respectively. The introns found associated with the CPIJ017592 gene are also much larger, leading to a nearly fourfold increase in size when compared with the CPIJ017593 gene. Genes CPIJ007581 and CPIJ007582 are also localized next to each other and are organized similarly to the CPIJ017592/CPIJ017593 pair, possibly representing a duplication event. CPIJ007581 is more closely related to CPIJ017592, with the two proteins mainly differing in their N-terminuses. In contrast, CPIJ007582 is equivalent to CPIJ017593 but with a longer C-terminus. Two other pantetheinase genes, CPIJ007583 and CPIJ007584, are positioned after the CPIJ007581/CPIJ007582 pair and both are more divergent in sequence, but with overall identity greater with the CPIJ017593/CPIJ007582 genes and with a similar exon/intron organization.

### 3.5. Alignment of Cx. quinquefasciatus Pantetheinase Sequences and Homology Analyses

The predicted proteins encoded by the *Cx. quinquefasciatus* pantetheinase genes range in length from 494 to 555 amino acids. To further understand the diversity between them, a sequence alignment was carried out comparing the six pantetheinases identified (Figure 5). Differences between the various proteins are distributed throughout their sequences, with more relevant differences identified within specific N- and C-terminal ends. Potential GPI anchor sites were investigated for all six proteins, with no sites predicted for the CPIJ017593/CPIJ007582 pair but with a likely GPI-modification site identified for CPIJ017592, within a motif generally conserved for CPIJ007581, and GPI anchor sites also found for both CPIJ007583 and CPIJ007584. These results might reflect differences in localization for the various proteins, either soluble or anchored to cellular membranes. Searches for potential glycosylation sites were also carried out, with multiple N-glycosylation and O-glycosylation sites predicted for nearly all the *Cx. quinquefasciatus* pantetheinases, with the only exception being CPIJ007583, where no possible O-glycosylation sites were found. Further searches for pantetheinase genes and orthologs in other representative and medically important vector species, such as *Aedes aegypti* and *Anopheles gambiae*, were performed. A much-reduced number of pantetheinases encoding genes were found in these species, at least based on the reference genome sequences available. For *Ae. aegypti*, three pantetheinase genes were found, with two of those most closely related to the *Cx. quinquefasciatus* CPIJ007584, while the third *Ae. aegypti* gene showed a greater but equivalent homology to the remaining five genes from *Cx. quinquefasciatus*. A similar profile was seen for *An. gambiae*, where only two pantetheinase genes were found, one more closely related to CPIJ007584, while the second was equally related to the remaining five *Cx. quinquefasciatus* genes. Searches using the *Drosophila melanogaster* genome were also performed with five pantetheinase genes identified, but all with equivalent homologies to the different *Cx. quinquefasciatus* genes, indicating distinct gene duplication events. These results are compatible with a specific expansion in the number of pantetheinase genes from *Cx. quinquefasciatus,* which is not found in the other culicid species investigated here.

### 3.6. Expression Analysis of a Recombinant Pantetheinase

To continue the investigation of the CPIJ017593 pantetheinase, its gene was first cloned and expressed in *Escherichia coli* with an added, C-terminal, poly-histidine tag and missing its C-terminal end. The resulting, affinity-purified, recombinant protein migrated under denaturing conditions with a molecular weight of approximately 70 kDa (Figure 6A, left panel), contrasting with the 57 kDa molecular weight predicted according to its amino acid sequence. Immunodetection using a monoclonal anti-His antibody recognized this recombinant protein (Figure 6A, center) and the mass spectrometry analysis of the purified band identified unique peptides compatible with the *Cx. quinquefasciatus* CPIJ017593, confirming its identity. Greater amounts of this recombinant protein were then expressed in *E. coli* and used to produce polyclonal antibodies. Affinity-purified anti-pantetheinase antibodies recovered from the rabbit antiserum were then validated by testing with the recombinant pantetheinase from *E. coli* (Figure 6A, right). The same truncated fragment from the CPIJ017593 gene was also used to express the corresponding protein in insect *Sf*9 cells. After affinity purification, the his-tagged protein from the culture media was directly visualized through Coomassie staining of denaturing gels and its identity confirmed through immunodetection with both anti-His and anti-pantetheinase antibodies (Figure 6B). This ~70 kDa band was also subjected to mass spectrometry that confirmed its identity as the *Cx. quinquefasciatus* CPIJ017593 pantetheinase. To investigate whether the 70 kDa pantetheinase was targeted in vivo by N-glycosylation, and considering the presence of three predicted N-glycosylation sites found in its sequence, the recombinant protein purified from *Sf*9 cells was treated with the endoglycosidase PNGase F. The treated pantetheinase showed a discrete reduction in molecular weight of 1–2 kDa when compared with the untreated control protein (Figure 6C). Similar results were seen when the treatment was performed on the nondenatured recombinant proteins. For this assay, we used as positive control the recombinant Aam1 protein also produced in *Sf*9 cells, previously reported to be N-glycosylated [43]. Upon treatment with PNGase F, Aam1 showed the expected decrease in molecular weight, from ~73 to ~66 kDa. These experiments indicate that minor, if any, glycosylation events are directed to the protein expressed in *Sf*9 cells and, presumably, to the native protein from *Cx. quinquefasciatus*.

### 3.7. Pantetheinase Transcript Abundance in Individual Larvae

Aiming to determine its relative abundance in individual larvae, the relative quantification of the CPIJ017593 pantetheinase transcript was investigated by RT-qPCR assays in individual larvae from the two Bin-resistant colonies and compared to larvae susceptible to the toxin (S). Levels of the pantetheinase transcript in individual S larvae showed some variation among individuals but, in general, were comparable to the profile of the one susceptible larva sample used as reference. In contrast, almost all REC and REC-2 resistant larvae showed a significantly reduced pantetheinase abundance profile compared to the susceptible reference larva (Figure 7A), which agrees with the transcriptomic data. The average relative quantification in log2FC of all individuals was lower in the REC (−2.91 ± 2.27) and REC-2 (−2.03 ± 1.55) resistant larvae compared to those from the susceptible colony, whose transcripts in most larvae were more abundant within a wide range (0.76 ± 1.79). These results were statistically significant for the comparison between the S colony versus REC (t_(8)_ = 69.35; *p* < 0.0001) and REC 2 (t_(7.2)_ = 73.40; *p <* 0.0001) colonies. Nevertheless, there was no statistical difference in average transcript quantification between the resistant colonies (t_(1.9)_ = 65.68; *p* = 0.0621). The abundance of the *18S* endogenous control transcript was similar in larvae from both colonies, with the full data from this analysis presented in Appendix A.

The availability of the anti-pantetheinase antibodies raised the possibility of investigating the expression of the native protein in the insect larvae. Indeed, a ~70 kDa protein was recognized by the anti-Pan antibodies in the BBMF samples (Figure 7B). These antibodies also recognized other proteins, in particular a 45 kDa protein, which was often co-detected in assays using recombinant proteins, suggesting that it could be a degraded form of pantetheinase (Appendix A). These results confirm the expression of at least one pantetheinase as a ~70 kDa isoform in the membrane fractions of the insect larvae midgut.

## 4. Discussion

The resistance to the Bin toxin has been previously solely attributed to the presence of various *cqm1* mutant alleles in homozygous larvae that are responsible for the lack of expression of the midgut-bound Cqm1 receptor [5,21,25]. Taken together, data from our previous findings [26,27] and this study, however, suggest that, in addition to the downregulation of the resistant *cqm1* alleles, the differential expression of other proteins might be important for the resistance phenotype. The differential expression of some transcripts, including those for cqm1 and pantetheinase, was validated by RT-qPCR, although some level of nonspecific detection could be possible considering the primers used (Appendix A). The pantetheinase downregulation can then be considered as an important marker for the monitoring of Bin resistance. When the pantetheinase transcripts were quantified in individual susceptible larvae, a variable profile in their levels was observed, suggesting a high diversity among individuals, but nevertheless nearly all Bin-resistant larvae showed a much lower abundance for these transcripts. Specific roles associated with Bin-toxin activity and/or resistance remain unknown not only for pantetheinase, but also for most other proteins whose transcripts were differentially expressed in the resistant larvae, with the exception of *cqm1*. Noteworthy is the specific set of downregulated and upregulated genes recorded here that are involved in insect growth and metamorphosis and whose potential relation with the Bin toxin mode of action deserves further investigation. An association between toxicity to insects with growth has been reported, for instance, for spinosad, a biological larvicide based on metabolites of the bacterium *Saccharopolyspora spinosa* that acts as a neurotoxin [44] which has been recently investigated for its potential effects as an insect growth regulator [45,46]. DEGs possibly related to the toxin mode of action could also be those encoding for regulators of DNA damage, as evidence for the involvement of apoptosis in the mode of action of Bin and Cry toxins has been shown [47,48,49]. Indeed, the transcriptome of the RIAB59 resistant larvae also pointed out several DEGs related to apoptosis and DNA maintenance [26]. A more in-depth investigation of other DEGs identified here would certainly suggest other important molecules and pathways that could be investigated in those Bin-resistant individuals [50].

Concerning the pantetheinases or vanin-like proteins, the focus of this study, to date no proper characterization of these enzyme in mosquitoes has been carried out. Indeed, only a few reports on pantetheinases from insects have emerged based on proteomic studies. These proteins have been detected alongside others found in the silk glands from *Bombix mori* [51], in the peritrophic membrane of *Mamestra configurata* [52], and in the midgut microvillar proteins of *Spodoptera frugiperda* [53,54]. The immunodetection shown in the present study, in samples from the midgut brush border membranes of *Cx. quinquefasciatus* larvae, is consistent with the expression of one or more pantetheinases as membrane-associated proteins in this species. The human vanin 1 and vanin 2 proteins are bound to the cell membrane by a GPI anchor [55,56] and this implies a possible binding of the mosquito pantetheinases to the midgut membrane also through GPI anchors. Vanin 1 and vanin 2 were also found to be glycosylated with differences between predicted (57 kDa) and observed (70 kDa) molecular weight being fully attributed to the presence of N-glycosides [57,58]. Here, despite the presence of potential N-glycosylation sites, at maximum only a discrete presence of N-glycosides can be expected based on the data with the recombinant pantetheinase. Regarding its expression in *Cx. quinquefasciatus* larvae as a soluble or membrane protein, the in silico prediction did not indicate a GPI anchor for the specific sequence investigated. However, the first detection of this pantetheinase in *Cx. quinquefasciatus* larvae [27], and the immunodetection shown in the present study in midgut brush border membrane samples, is compatible with its expression as a membrane-bound protein. Other pantetheinases detected in insects were found as midgut microvillar proteins [52,53,54], as well as the human vanin 1 and vanin 2 proteins [55,56]. As for the discrepancy in size observed for the recombinant *Cx. quinquefasciatus* pantetheinase, it could be attributed, for instance, to folding properties or to anomalous SDS-PAGE migration due to interactions with detergents [59,60,61].

In contrast to the well-established role for the Cqm1 receptor in mediating the action of the Bin-toxin [16,23], and which explains the downregulation seen here for *cqm1* in the Bin-resistant colonies, the strong pantetheinase downregulation observed highlights the need to better understand its association with the resistance phenotype. Thus far, there are mostly no data on the involvement of pantetheinases in the mode of action of microbial insecticidal proteins. Studies comparing the gut expression profiles of two strains of the lepidopteran *Ostrinia nubialis*, susceptible and resistant to the Cry1Ac toxin, showed differential expression of vanin-like proteins upon exposure to the toxin, but no correlation between the resistance phenotype and specific changes in expression was observed, and these proteins were not investigated further [62,63]. The characterization studies of the midgut receptors for Cry toxins active to lepidopteran and dipteran larva, for instance, first identified cadherins, aminopeptidases, and alkaline phosphatases [8,64,65]. More recently, however, other molecules, such as ABC and ATP-binding cassette transporters, have been implied in their toxicity mechanism [66,67,68,69,70]. Further investigation is therefore necessary to evaluate any potential involvement of the *Cx. quinquefasciatus* pantetheinase in the Bin toxin mode of action. An alternative possibility for the observed downregulation of the pantetheinase transcripts in the Bin-resistant larvae would be for this to result from an independent co-selection event induced by the exposure to the toxin.

The primary function of pantetheinases is to catalyze the conversion of pantetheine into pantethonate (vitamin B5) and cysteamine, as reviewed by Kaskow et al. [71]. Those two products are involved in the metabolism of fatty acids and have also been implicated in the oxidative stress and in inflammation cycles which are related to the liberation of cysteamine [72]. Regarding the metabolism of fatty acids in humans, for instance, the *vnn1* gene is regulated by the peroxisome proliferator-activated receptor alpha (PPAR-α), which is involved in fat oxidation as an energy source [73,74,75]. Another study demonstrated the major role for pantetheinases in lipolysis, due to its capacity to induce the activity of the peroxisome proliferator-activated receptor gamma (PPAR-γ), which is an activator of promoters of lipolytic genes [76]. Indeed, the Bin-resistant larvae from the RIAB59 colony showed altered transcription levels of several genes associated with lipid and carbohydrate metabolism [26], as well as altered energy reserves [50]. Regarding immunity, the coenzyme A (CoA), which is involved in the production of pantetheine being restored using pantethonate, is a critical molecule for the development of *Plasmodium* in humans and in mosquitoes [77]. Parasites seem to rely on the availability of exogenous pantethonate for CoA production, and its depletion in *Anopheles stephensi* was associated with the reduction of infection [78]. These findings offer interesting perspectives for the investigation on physiological roles for pantetheinases in mosquitoes, exploring their potential involvement in metabolism, immunity, and homeostasis and which could potentially help to explain the changes in expression seen here with the resistance larvae. A major question that is raised by this work deals with the mechanisms behind the simultaneous downregulation of the pantetheinase transcripts as well as those encoding the various differentially expressed genes identified here. Downregulation of the *cqm1* transcript is associated with frameshift mutations within its gene. These introduce premature stop codons within the mRNA that localize before the last exon-junction. It has been postulated then that a non-sense-mediated decay mechanism might be associated with the mRNA degradation and could potentially explain the *cqm1* downregulation seen for the Bin-resistant larvae [5]. A similar mechanism targeting the multiple pantetheinase transcripts also seen to be downregulated would require each corresponding gene to be mutated, a very unlikely possibility. New alternative mechanisms are therefore expected to be involved and will need to be investigated.

## 5. Conclusions

Our study identified several transcripts from *Cx. quinquefasciatus* that are differentially regulated in three different colonies resistant to the *L. sphaericus* Bin toxin. Transcripts encoding the pantetheinase enzyme are consistently downregulated and potentially represent novel Bin toxin resistance markers. Pantheteinase is likely to localize to the epithelium membrane from the larvae midgut, similar to the Cqm1 receptor, as a 70 kDa protein that does not seem to be extensively modified by glycosylation events. The contrasting transcriptomic profile of pantetheinase observed between susceptible and Bin-resistant larvae opens up perspectives to investigate key aspects of this enzyme in mosquitoes further, including in their metabolism and immunity.

## Figures and Tables

**Figure 1 biomolecules-14-00033-f001:**
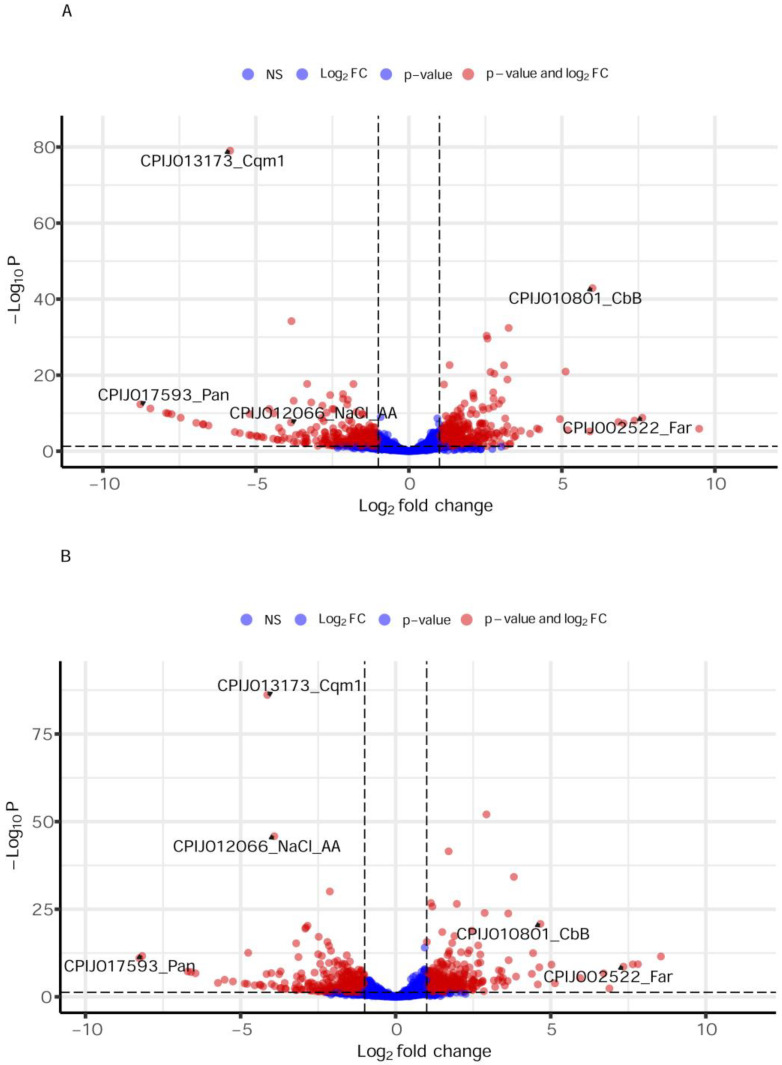
Volcano plots representing the genes found in larvae from two Bin-resistant *Culex quinquefasciatus* colonies (REC *n*= 5633, REC-2 *n*= 6434), compared to larvae from a susceptible colony (S). The plots were derived from RNA-seq and show the correlation between genes with log2 fold change (log2 FC) values ≥ 1 and FDR-corrected *p-*values ≤ 0.05. Genes with log2 FC values ≥ 1 and the respective *p-*values ≤ 0.05 are represent in red and listed in Appendix A. Genes displaying only one cut off parameter, or genes that were not significant (NS), are represented in blue. (**A**) REC × S comparison. (**B**) REC-2 × S comparison. Genes that were subjected to the validation shown are indicated: Cqm1-*Culex quinquefasciatus* maltase 1; Pan-Pantetheinase; NaCl sodium/chloride-dependent amino acid transporter putative; CbB-Carboxypeptidase B precursor; Far-Farnesol dehydrogenase.

**Figure 2 biomolecules-14-00033-f002:**
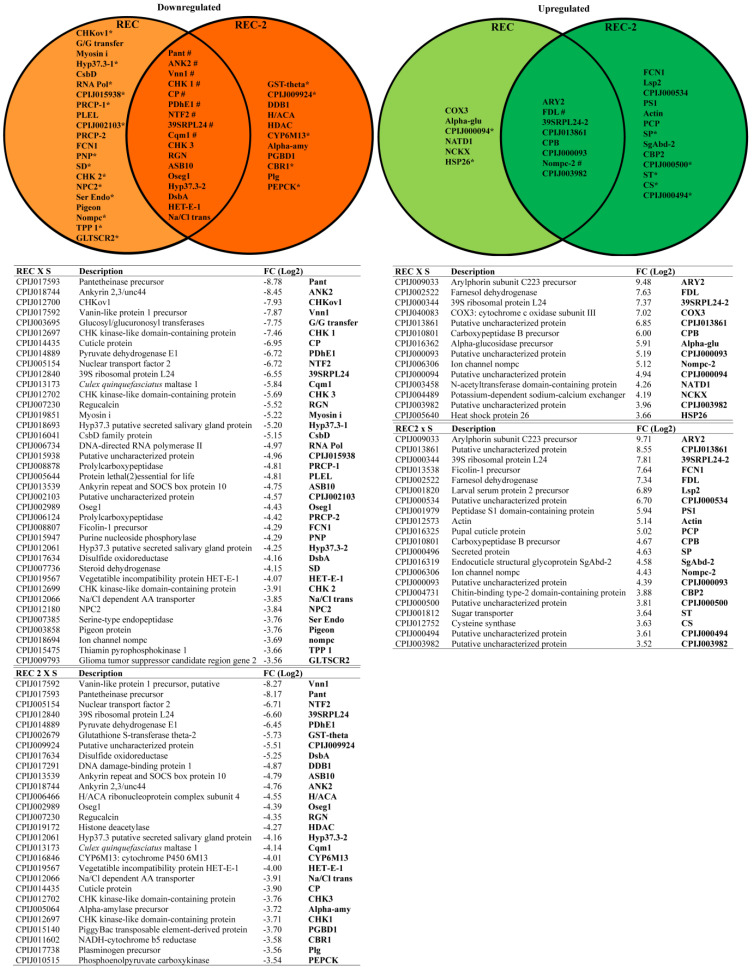
Differentially expressed genes found in two colonies of Bin-resistant *Culex quinquefasciatus* larvae (REC and REC-2). The Venn diagrams show a direct comparison of the most downregulated or upregulated transcripts common to both colonies when compared to the susceptible (S) colony. Those genes are derived from Appendix A larvae but have a log2 fold change (log2FC) ≥ 3.5 and FDR-corrected *p-*values ≤ 0.05, in at least one of the two colonies. (*) Transcripts having a log2FC ≥ 3.5 in one colony but with a log2FC in the second colony ≤ 3.5 but ≥ 1. (#) Transcripts also found to be differentially expressed in the RIAB59 colony with a log2FC ≥ 3.5, as previously reported [26]. The tables below detail the transcripts shown in the diagrams.

**Figure 3 biomolecules-14-00033-f003:**
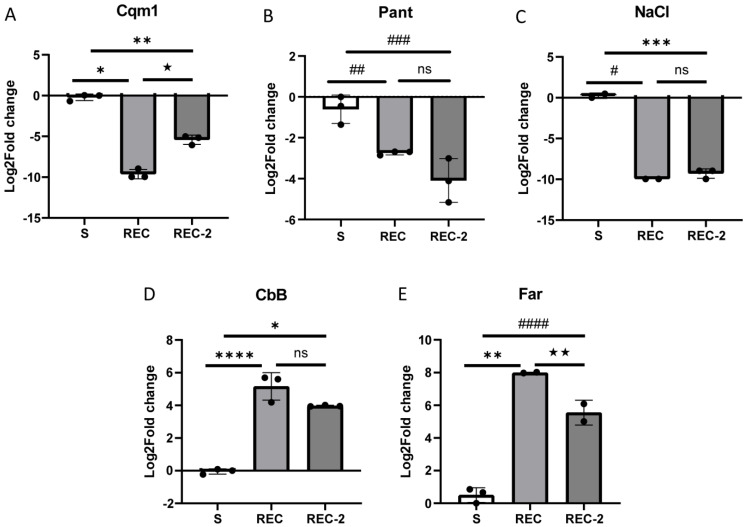
Relative quantification using RT-qPCR of the *Culex quinquefasciatus* transcripts from the Bin-resistant larvae (REC and REC-2) in comparison with those from the susceptible larvae (S). (**A**) *cqm1* (Cqm1 CPIJ013173). (**B**) *pantetheinase* (Pan CPIJ017593). (**C**) *sodium/chloride dependent amino acid transporter putative* (NaCl CPIJ012066). (**D**) *carboxypeptidase B precursor* (CbB CPIJ010801). (**E**) *farnesol dehydrogenase* (Far CPIJ002522). Gene expression levels are relative to those from the endogenous control *18S* gene used for normalization. Each column represents the mean with SEM obtained from three biological replicates from each colony. All values considered statistically different had *p-*values < 0.05 (* *p* = 0.0001, ** *p* = 0.0002, *** *p* = 0.0003, **** *p* = 0.0005, ^#^ *p* = 0.0006, ^##^ *p* = 0.0061, ^###^ *p* = 0.0091, ^####^ *p* = 0.0024, ^★^ *p* = 0.0008, ^★★^ *p* = 0.0454), or they were not significant (ns).

**Figure 4 biomolecules-14-00033-f004:**
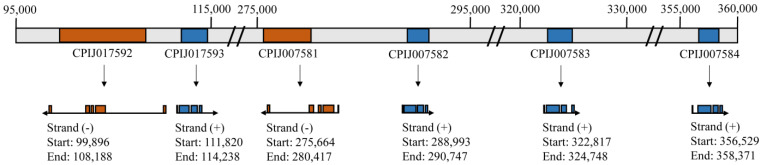
Organization of the pantetheinase genes annotated within the *Culex quinquefasciatus* genome. The scheme summarizes the genome data available at the VectorBase database (https://www.vectorbase.org accessed on 20 September 2023) for the two genome regions encoding the contiguous sets of six pantetheinase genes. The exon/intron organization for the corresponding coding sequences, the transcription start sites, and orientation of the various genes are indicated.

**Figure 5 biomolecules-14-00033-f005:**
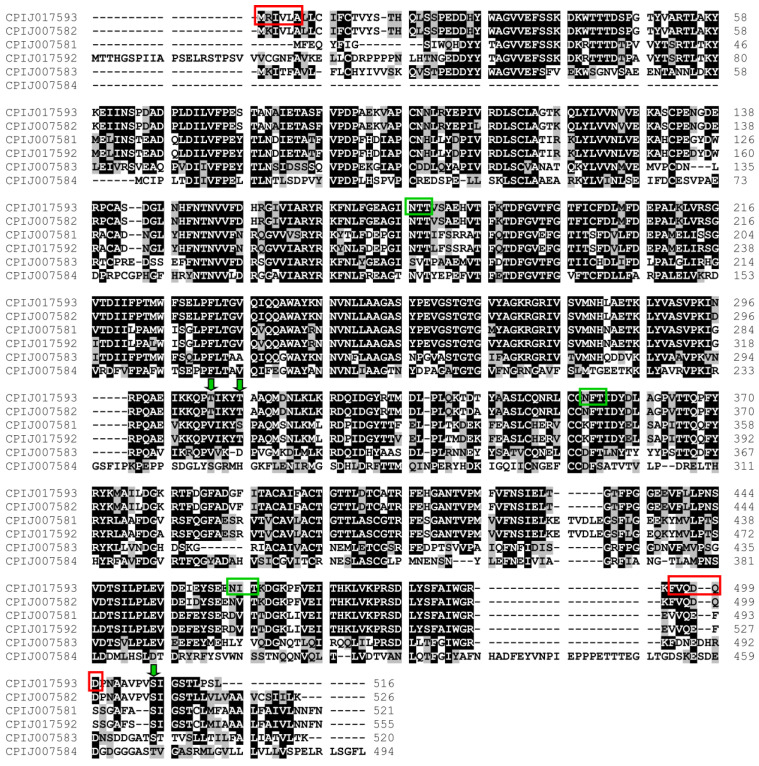
Amino acid sequence alignment of the *Culex quinquefasciatus* pantetheinases available at Vectorbase (www.vectobase.org accessed on 20 September 2023). Identical and similar amino acids are shown in black and gray, respectively. Primers used for gene amplification (red boxes), N-glycosylation (green boxes), and O-glycosylation sites (arrows) are indicated in the CPIJ017593 sequence.

**Figure 6 biomolecules-14-00033-f006:**
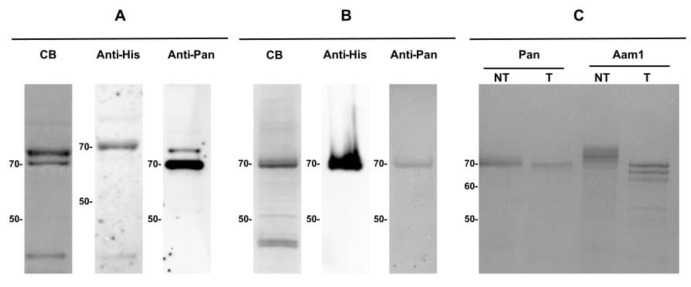
Recombinant expression of the pantetheinase CPIJ017593 from *Culex quinquefasciatus* and investigation of glycosylation events. (**A**) Recombinant protein expressed *Escherichia coli*. (**B**) The same protein expressed in *Sf*9 cells. For both systems, the proteins were purified with NI-NTA resin and visualized in 10% SDS-PAGE gels stained with Coomassie blue (CB). They were also detected through immunoblotting using monoclonal antibodies directed to the poly-hystidine tag (anti-His) or polyclonal antibodies raised against the recombinant pantetheinase (anti-Pan). (**C**) Endoglycosidase treatment of recombinant proteins from *Sf*9 cells: pantetheinase (Pan) and an α-glucosidase from *Aedes aegypti* (Aam1). Samples were incubated with the enzyme PNGase F at 37 °C during 4 h, separated on 10% SDS-PAGE, and visualized with CB. NT. Incubation without the enzyme. T. Incubation with the enzyme. Molecular weight markers (kDa) are on the left. Original Western blot images can be found in the Appendix A.

**Figure 7 biomolecules-14-00033-f007:**
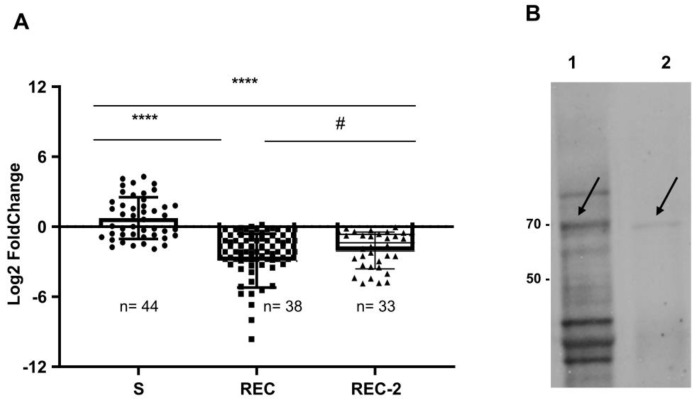
Abundance profile of the selected pantetheinase transcript (CPIJ017593) in *Culex quinquefasciatus* larvae. (**A**) Relative quantification of transcripts in individual larvae from the susceptible (S) colony compared to the Bin-resistant ones (REC, REC-2), by qRT-PCR. Each column represents the mean with SEM obtained from three biological replicates from each colony. Statistically significant differences (**** *p* < 0.001), or not (# *p* = 0.06), are indicated. (**B**) Immunodetection of proteins from brush border membranes preparations (BBMF) from susceptible larvae using the polyclonal antibodies raised against the recombinant CPIJ017593 pantetheinase. 1. BBMF prepared using whole larvae. 2. BBMF prepared using dissected midguts. Molecular weight markers (kDa) on the left. Arrows show the ~70 kDa proteins. Original Western blot images can be found in the Appendix A.

## Data Availability

Raw data set is available in Appendix A; the transcriptome dataset (BioProject: PRJNA103356) is available at NCBI https://www.ncbi.nlm.nih.gov/sra/PRJNA1033561.

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
