# Peer review of "Culex quinquefasciatus Resistant to the Binary Toxin from Lysinibacillus sphaericus Displays a Consistent Downregulation of Pantetheinase Transcripts"

_biomolecules, 2023, doi:10.3390/biom14010033_

Round 1

Reviewer 1 Report

Comments and Suggestions for Authors

The article is very interesting and well written and the overall quality of the manuscript is excellent. The topic is very important because adding data regarding the evidence of new possible mechanisms of acquisition of resistance is a priority due to the risks associated with the resurgence of mosquito-borne diseases.  I do not detect any particular aspect that requires improvement or revision.

I have only recognized a repetition of the word "larvae" at line 562

I congratulate the Authors and wish them all the best for the future

Author Response

Manuscript ID: biomolecules-2665646

Answers to Reviewer 1

Comment. I have only recognized a repetition of the word "larvae" at line 562.

Answer. Thanks this was corrected. As request by other reviewers, changes and improvement of the R1 version were done.

Reviewer 2 Report

Comments and Suggestions for Authors

In this study, Rezende et al. tried to identify additional targets associated with Bin toxin resistance in Culex quinquefasciatus larvae midguts.

In transcriptomic analysis, DEGs were identified in midguts of larvae from two Bin-resistant Culex quinquefasciatus colonies.

The authors tried to validate the results of RNA-Seq using suboptimal primers.

After pantetheinase genes in silico analysis, the selected pantetheinase CPIJ017593 was cloned and studied in vitro for glycosylation and transcript levels.

There are several critical issues that must be resolved by the authors in the present version of the Manuscript.

Major issues

[1]

[1a]

Most statements on the role of pantetheinase transcript levels in Bin toxin resistance are excessive and must be omitted or loosened.

Please improve the Abstract and other sections of the manuscript.

[1b]

The major goal of this study is formulated in a fuzzy way.

[LL86-87]

 “…to investigate the status of pantetheinase in the transcriptome…”

Please clarify.

[2]

qPCR assay was done in a suboptimal way and is leaky.

[2a]

Suboptimal primers were applied for detection of cqm1, pan and cbB targets.

Please state the reasons to use suboptimal primers for detection of the target transcripts.

[2b]

CPIJ002522 (ser) primers are for farnesol dehydrogenase, not for serine 3-dehydrogenase.

Please clarify the reason to use these primers for detection.

[2c]

Amplicon lengths are incorrect for CPIJ012066, CPIJ010801, CPIJ002522 targets.

Please verify.

[3]

Data reported on the Fig 7 are inconsistent.

[3a]

The data depicted on the Fig 7 do not correspond to the data released in the Table S6

Please check whether Log2 or Log10 changes were depicted on the Fig. 7.

[3b]

The distribution of datapoints for the control group is strongly bimodal, biasing these results.

Please report the full technical details of the qPCR assay, including positions of samples in the wells (F2, C10, etc) and other issues that might influence the qPCR results.

Please also provide a short explanation for the FIG 7 bimodal distribution in the Discussion section.

[4]

The DEG transcriptomic analysis reporting is missed and must be improved extensively.

[4a]

The subsection 3.1 “Transcriptomic profile of Bin-resistant Cx. quinquefasciatus colonies” must be moved to the “Materials and Methods” section and extended.

[4b]

The pipeline utilized for the DEG transcriptomic analysis must be described in the “Materials and Methods” section in full details, with all utilities used and all parameters applied.

[4c]

For DEG results, please report all Volcano plots as supplementary figures.

Please compare Volcano plot results with the data reported as PCA.

Please discuss Volcano plots and PCA results, including the consistency of these results.

[4d]

The conventional Gene ontology analysis using the DEGs identified must be performed and reported.

Please use DAVID utility for GO analysis (https://david.ncifcrf.gov/).

Please report details on overrepresented entities, including Protein_Domains, GO:MF and GO:CC terms.

[5]

The data reporting is scarce and must be improved.

In particular, no detailed statistical reports supporting each finding were provided by the authors.

PCA and Fig 2 findings are lacking raw data.

[5a]

The authors must provide all raw data and all statistical calculations underlying results, reporting these in a form of a Supplementary table file (XLS or other).

[5b]

Where relevant in the main text of the manuscript, please provide exact values for both significant and non-significant P values.

For ANOVAs, please provide F values and degrees of freedom.

For example, one-way ANOVA results should be reported like F(1, 510) = 6.71, p = 0.009

For t-tests, please provide t-values and degrees of freedom.

For example, Welch's t test should be reported as follows t(33) = 3.87, p = 0.007

For other examples, please see https://www.bachelorprint.eu/apa-style/reporting-statistics-in-apa/.

[5c]

When needed, the data should be analysed by the two-way ANOVA with relevant factors assessed individually.

[5d]

The panels of the figure 2 have been provided without individual datapoints depicted with bars.

Please improve pictures, using jitter plot design in pair with bars.

Comments on the Quality of English Language

Language issues

[6]

Language is fuzzy and reader-unfriendly in a number of sentences.

[6a]

In a large number of sentences, the authors used incomplete clauses or fuzzy word orders.

Please improve.

Several examples are listed below.

[LL33-34]

“…larvicides have as their main active ingredient, toxins…” – please improve word order

[L47]

“…to the membrane bound Cyt1Aa (?),” – please clarify

[LL50-51]

“…larvicides producing crystals … play an important role in integrated control programs…” – Please clarify

[LL53-54]

“…remains a major concern, leading to resistance levels…” – please clarify

 [LL19-20]

“Further quantification of these transcripts confirmed a consistent downregulation (?) in Bin-resistant larvae.” – Please clarify

The abovementioned list is not exhaustive.

The authors must improve the text greatly, including the Introduction section

[6b]

The text is rich with mistypes that must be corrected.

A single example is “ribossomal”

Author Response

Manuscript ID: biomolecules-2665646

Answers to Reviewer 2

General  Comment - There are several critical issues that must be resolved by the

authors in the present version of the Manuscript.

Answer. We acknowledge the questions and corrections raised to improve the manuscript. A major review was done and changes are tracked in the manuscript.

Major issues

Comment [1] [1a]Most statements on the role of pantetheinase transcript levels in Bin toxin resistance are excessive and must be omitted or loosened. Please improve the Abstract and other sections of the manuscript.

Answer. The comments related to the role of pantetheinase transcript levels in Bin toxin resistance were revised to tonedown this aspect. However, the article showed a relation between the outstanding pantetheinase downregulation in two different Bin-resistant colonies compared to a susceptible, this finding and the possible reasons behind this association were kept.

Comment [1b] The major goal of this study is formulated in a fuzzy way. [LL86-87] “…to investigate the status of pantetheinase in the transcriptome…”Please clarify.

 Answer. This sentence changed toTherefore, the major goal of this study was to compare the level of transcription of the pantetheinase in larvae of two Cx quinquefasciatus Bin-resistant colonies (REC, REC-2) and to perform its in vitro identification through the recombinant expression in heterologous system”.

[2] qPCR assay was done in a suboptimal way and is leaky.

Comment [2a] Suboptimal primers were applied for the detection of cqm1, pan and cbB targets. Please state the reasons to use suboptimal primers for detection of the target transcripts.

Answer.  The purpose was not using suboptimal primers. The primers were designed using Primer Select (DNAStar) considering criteria to fine performance as dimers formation, hairspins, G/C contents. The meting temperature and annealing of primers were also verified using the Tm calculator (Thermo Fisher) and pilot assays were performed to check the reaction conditions. In this revision we analysed again and, indeed, blast aligment showed some reduction of specificity when either primer F or R were tested. It is likely that this could be partially due to genome updates that took place after the primers and assays were done. However, they still display a suitable condition to annealing and to our specific targets and the results of these relative quantitative by RT-qPCR obtained corroborate the differential transcription profile of RNAseq and thus the validation purpose.

Comment [2b] CPIJ002522 (ser) primers are for farnesol dehydrogenase, not for serine 3-dehydrogenase. Please clarify the reason to use these primers for detection.

Answer. The CPIJ002522 is annotated as “serine 3-dehydrogenase” in the Vectorbase, the reason why it was used. In the R1 version it was updated to farnesol dehydrogenase, based in annotation from Genebank, as suggested.

Comment [2c] Amplicon lengths are incorrect for CPIJ012066, CPIJ010801, CPIJ002522 targets.

Please verify.

Answer. This error derived from a typing mistake that change the frame in the Table lines. All primers and their respective amplicons lengths were revised and corrected in Table S1.

[3] Data reported on the Fig 7 are inconsistent.

Comment [3a] The data depicted on the Fig 7 do not correspond to the data released in the Table S6. Please check whether Log2 or Log10 changes were depicted on the Fig. 7.

Answer. We detected this mistake, indeed in the previous version Log10 fold changes values were shown. This was corrected and, in the R1 version, data is expressed in log2 fold changes, comprising figure, text and the respective supplementary table.

Comment [3b] The distribution of datapoints for the control group is strongly bimodal, biasing these results. Please report the full technical details of the qPCR assay, including positions of samples in the wells (F2, C10, etc) and other issues that might influence the qPCR results. Please also provide a short explanation for the FIG 7 bimodal distribution in the Discussion section.

Answer. The quantification of this transcript showed a variable profile compared to the susceptible larvae sample that was used as the reference in all assays. This result was expected since individual larvae samples were analyzed and not pools of larvae. In previous studies of our team (Araujo et al 2013doi 10.1186/1756-3305-6-297, Menezes et al 2021 10.1002/ps.6349, Menezes et al 2023 10.1186/s13071-023-05893-z), revealing a large variation assessing the catalytic activities of some enzymes also testing individual larvae variation instead of larvae pools. The quantification of transcripts itself is also subjected to great variations due to several factors. For panthetheinase so far, transcription and expression data, and other information to better explain such variation is not available. The profile seen in our analysis display a clear difference between the profile of most resistant individuals and that of susceptible larvae, whose differences were statistically significant and demonstrated that resistant individual have a downregulation of this enzyme. To address this aspect a comment was inserted in Discussion (R1, L525-528).

Comment [4] The DEG transcriptomic analysis reporting is missed and must be

improved extensively.

Answer. A major revision was done regarding the transcriptome dataset. The genes sets were analyzed again using a more recent version of DESeq2 (1.40.2). This provided a new list of DEGs which is now presented. It worth noting that log2 fold change values were greater than the previous analysis but the strong downregulation of the pantetheinases were kept, they were the topmost downregulated transcripts in both Bin-resistant colonies. It is important to notice that, in this manuscript, it was our intention to do not fully exploit the transcriptome since the pantetheinase was the focus while other aspects are under investigation.

.

Comment [4a] The subsection 3.1 “Transcriptomic profile of Bin-resistant Cx. quinquefasciatus colonies” must be moved to the “Materials and Methods” section and extended.

Answer. We did not understand this request, as the subsection 3.1 presents the results of the transcriptome. Altought these include an overview analysis of the dataset obtaind the authors think it would be better to keep this data in the results section. Addition information was also included as requestes

Comment [4b] The pipeline utilized for the DEG transcriptomic analysis must be described in the “Materials and Methods” section in full details, with all utilities used and all parameters applied.

Answer. The information is available in Rezende et al 2019 and it was described here as requested. Other methodological information were added.

Comment [4c] For DEG results, please report all Volcano plots as supplementary figures.

Please compare Volcano plot results with the data reported as PCA. Please discuss Volcano plots and PCA results, including the consistency of these results.

Answer. As informed above Volcano plots were reported and comments were introduced. We evaluated that these plots should be shown as a major figure in the manuscript.

Comment [4d] The conventional Gene ontology analysis using the DEGs identified must be performed and reported. Please use DAVID utility for GO analysis  https://david.ncifcrf.gov/). Please report details on overrepresented entities, including Protein_Domains, GO:MF and GO:CC terms.

Answer. Yes, the gene ontology using DAVID utility was included as supplementary material for information of enrichment of those terms, and the profile of each resistant colony compared with the reference colony was presented in the text.

[5] The data reporting is scarce and must be improved. In particular, no detailed statistical reports supporting each finding were provided by the authors.

PCA and Fig 2 findings are lacking raw data.

Comment [5a] The authors must provide all raw data and all statistical calculations underlying results, reporting these in a form of a Supplementary table file (XLS or other).

Answer. Statistical reports are now presented in supplementary Table S7 and other additional dataset was provided in Table S3 (PCA Statistics), Table S5 (GO and Interpro dataset by DAVID tool), Table S6 (RNA-seq validation), TableS7 (Statistics RT-qPCR)), Table S9 (Dataset RT-qPCR pantetheinases).

Comment [5b] Where relevant in the main text of the manuscript, please provide exact values for both significant and non-significant P values. For ANOVAs, please provide F values and degrees of freedom. For example, one-way ANOVA results should be reported like F(1,510) = 6.71, p = 0.009 For t-tests, please provide t-values and degrees of freedom.

For example, Welch's t test should be reported as follows t(33) = 3.87, p = 0.007. For other examples, please see https://www.bachelorprint.eu/apastyle/reporting-statistics-in-apa/.

Answer. Statistical reports are available in Table S10 and results are also reported in the text.

Comment [5c] When needed, the data should be analyzed by the two-way ANOVA with relevant factors assessed individually.

Answer. In this manuscript our goal was to compare each resistant colony with the refence one and, for this reason only the T-test was used and not ANOVA.

Comment [5d] The panels of the figure 2 have been provided without individual datapoints depicted with bars. Please improve pictures, using jitter plot design in pair with bars.

Answer. This figure, now Figure 3 was corrected as requested.

Comments on the Quality of English Language

Language issues

[6] Language is fuzzy and reader-unfriendly in a number of sentences.

Comment [6a] In a large number of sentences, the authors used incomplete clauses or fuzzy word orders. Please improve. Several examples are listed below.

Answer. We tried as much as possible to improve some sentences, thanks for the examples below. The manuscript-R! was also subjected to a review in order improve the quality of English language.

Comment [LL33-34] “…larvicides have as their main active ingredient, toxins…” –

please improve word order

Answer. The sentence was modified.

Comment [L47] “…to the membrane bound Cyt1Aa (?),” – please clarify

Answer. The information was clarified.

Comment [LL50-51] “…larvicides producing crystals … play an important role in

integrated control programs…” – Please clarify.

Answer. The information was clarified.

Comment [LL53-54] “…remains a major concern, leading to resistance levels…” –

please clarify.

Answer. The information was clarified.

Comment [LL19-20] “Further quantification of these transcripts confirmed a consistent downregulation (?) in Bin-resistant larvae.” – Please clarify. The abovementioned list is not exhaustive.

The authors must improve the text greatly, including the Introduction section

Answer. This sentence was changed since our goal was to show that the panthetheines were the topmost downregulated genes in both Bin-resistant colonies.

Comment [6b]

The text is rich with mistypes that must be corrected.

A single example is “ribossomal”

Answer. This was corrected as well as other mistakes found, as much as possible during the review.

Reviewer 3 Report

Comments and Suggestions for Authors

Comments and suggestions for authors:

The study entitled “Culex Quinquefasciatus Resistant to The Binary Toxin From Lysinibacillus Sphaericus Displays A Consistent Downregulation of Pantetheinase Transcripts” authored by Tatiana Rezende, Heverly Suzany Menezes, Antonio Rezende, Milena Cavalcanti, Yuri Silva1, Osvaldo de-Melo-Neto, Tatiany Romão and Maria Helena Silva-Filha, is devoted to highlight the status of pantetheinase in the transcriptome of two Cx. quinquefasciatus Bin-resistant colonies. The presented research is of scientific importance and contains the potential for applied use. All the sections are well written and provided detailed information.

There is a minor question regarding selection of pantetheinase gene in line#156. What is the basis for selecting this gene CPIJ017593 and not others listed in supplementary table 4.

#723, #729, #735, #764, recheck scientific names.

Author Response

Manuscript ID: biomolecules-2665646

Answers to Reviewer 3

 Comment - There is a minor question regarding selection of pantetheinase gene in line#156. What is the basis for selecting this gene CPIJ017593 and not others listed in supplementary table 4.

Answer. The reason is explained in section 3.2 (R1, line 330-334). The CPIJ017593 transcript was the greatest repression among the pantheteinases: “The CPIJ017593 pantetheinase transcript showed the second greatest downregulation in the REC colony and was the most downregulated transcript among the REC-2 larvae. Another pantetheinase coding gene (CPIJ017592) was also detected among the most downregulated transcripts in both sets of Bin-resistant larvae”. The four others did not showed significant differential expression as shown in Table S8.

Comment - #723, #729, #735, #764, recheck scientific names.

Answer. The corrections were done.

Round 2

Reviewer 2 Report

Comments and Suggestions for Authors

 [1]

All reported primers must be validated via Blast service against transcripts specific for Culex quinquefasciatus.

In case of discrepancies between sequences of primers used in assays and sequences of transcripts identified via blast utility, all suboptimal primers must be clearly identified in the “Materials and Methods” section and in the Table S1.

[1a]

In particular, each primers sequence deviating from database reference sequences must be clearly indicated in the Table S1, compared with relevant database sequences.

[1b]

The possible limitation of this study (due to the use of suboptimal primers to specific targets) must be clearly indicated in the Discussion section.

Comments on the Quality of English Language

[2]
Language is still fuzzy, especially in the Abstract and in the Introduction sections.

In particular,
please avoid using complex clauses like “…protoxins … and which are …”
please avoid using of undetermined terms like “a previous transcriptome”

Author Response

Manuscript ID: biomolecules-2665646_R2

Answers to Reviewer 2_R2

We thank for the questions raised aiming to improving the manuscript and the answers are described below.

 [1] All reported primers must be validated via Blast service against transcripts specific for Culex quinquefasciatus.  In case of discrepancies between sequences of primers used in assays and sequences of transcripts identified via blast utility, all suboptimal primers must be clearly identified in the “Materials and Methods” section and in the Table S1.

 Query [1a]. In particular, each primers sequence deviating from database reference sequences must be clearly indicated in the Table S1, compared with relevant database sequences.

Answer. The Table S1 was revised and, for each primer, verification using the Blast service (option: transcripts specific for Culex quinquefasciatus), was provided. This information was given in the Methodology (lines 151-152). All primers target our sequence as the first hit, and we included the possible second hit for clarity.

Query [1b]. The possible limitation of this study (due to the use of suboptimal primers to specific targets) must be clearly indicated in the Discussion section.

Answer. As requested, a sentence was introduced in the discussion (lines 532-535) to inform that nonspecific detection of transcripts was possible.

Comments on the Quality of English Language

Query [2]. Language is still fuzzy, especially in the Abstract and in the Introduction sections.
In particular,
please avoid using complex clauses like “…protoxins … and which are …”
please avoid using of undetermined terms like “a previous transcriptome”.

Answer. We did the correction of several mistakes, and improved the flow of some sentences,  and the manuscript was revised by a fluent biology researcher. We hope the manuscript achieved an acceptable language level for publication.